# The Impact of Diffuse Idiopathic Skeletal Hyperostosis on Nutritional Status, Neurological Outcome, and Perioperative Complications in Patients with Cervical Spinal Cord Injury

**DOI:** 10.3390/jcm12175714

**Published:** 2023-09-01

**Authors:** Tomoaki Shimizu, Kota Suda, Satoko Matsumoto Harmon, Miki Komatsu, Masahiro Ota, Hiroki Ushirozako, Akio Minami, Masahiko Takahata, Norimasa Iwasaki, Hiroshi Takahashi, Masashi Yamazaki

**Affiliations:** 1Department of Orthopaedic Surgery, Hokkaido Spinal Cord Injury Center, 3-1 Higashi 4 Minami 1, Bibai 072-0015, Hokkaido, Japanverisa0808@gmail.com (H.U.);; 2Department of Orthopaedic Surgery, Hokkaido University Graduate School of Medicine, Sapporo 060-0815, Hokkaido, Japan; 3Department of Orthopaedic Surgery, Faculty of Medicine, University of Tsukuba, Tsukuba 305-8575, Ibaraki, Japan

**Keywords:** diffuse idiopathic skeletal hyperostosis, cervical spinal cord injury, nutritional status, perioperative complications, neurological outcomes

## Abstract

This retrospective study aimed to investigate the characteristics of patients with cervical spinal cord injuries (CSCI) with diffuse idiopathic skeletal hyperostosis (DISH). We included 153 consecutive patients with CSCI who underwent posterior decompression and fusion surgery. The patients were divided into two groups based on the presence of DISH. Patient characteristics, neurological status on admission, nutritional status, perioperative laboratory variables, complications, neurological outcomes at discharge, and medical costs were compared between the groups. The DISH group (n = 24) had significantly older patients (72.1 vs. 65.9, *p* = 0.036), more patients with low-impact trauma (62.5% vs. 34.1%, *p* = 0.009), and a lower preoperative prognostic nutritional index on admission (39.8 vs. 42.5, *p* = 0.014) than the non-DISH group (n =129). Patients with DISH had significantly higher rates of ventilator management (16.7% vs. 3.1%, *p* = 0.022) and pneumonia (29.2% vs. 8.5%, *p* = 0.010). There was no significant difference in medical costs and neurological outcomes on discharge. Patients with CSCI and DISH were older, had poor nutritional status, and were prone to postoperative respiratory complications, while no differences were found between the neurological outcomes of patients with CSCI with and without DISH.

## 1. Introduction

Diffuse idiopathic skeletal hyperostosis (DISH) is a noninflammatory condition in which the spinal longitudinal ligaments and entheses gradually ossify, reducing mobility in the affected section [1]. Because of secondary osteoporosis and multilevel bony vertebral fusions that produce long lever arms, even low-energy trauma can cause fractures with an increased risk of neurological injury. Fractures occur four times more frequently in the ankylosed than in the nonankylosed spine [2]. Therefore, early diagnosis of injury and rigid spinal fixation are usually recommended for DISH fractures. However, higher perioperative complications have been reported for DISH surgery [3,4].

Cervical spinal cord injury (CSCI) is a serious complication of traumatic cervical fracture of the ankylosed spine due to its morphological instability. Compared with fractures in other sections of the spine, cervical fractures are more likely to correlate with SCI [4,5]. The rate of SCI after traumatic cervical fractures in patients with an ankylosed spine is 44.0–86.8% [4,5,6,7]. Despite appropriate medical management, perioperative complication rates of traumatic SCI surgery are reportedly higher compared with those of other spinal disorders [8]. Due to decreased respiratory muscle strength, retention of secretions, and autonomic dysfunction, as many as two-thirds of patients with acute SCI develop pulmonary complications, such as atelectasis, pneumonia, and respiratory failure [9].

While various factors including patient background [10], surgical strategy [11,12], neurological status [13], and nutritional status [14] supposedly affect SCI perioperative complications and neurological prognosis, the impact of DISH on CSCI patients has not yet been described. This study aimed to investigate the baseline characteristics of CSCI patients with DISH and to determine how DISH affects nutritional status, neurological prognosis, and perioperative complications. We hypothesized that CSCI patients with DISH would have a high risk of perioperative complications and a poorer prognosis.

## 2. Materials and Methods

Patient selection

This retrospective analysis of patient data was approved by the institutional review board of our hospital (IRB No. 76). All procedures involving human participants were conducted in accordance with the Declaration of Helsinki. The requirement for consent was waived by the institutional review board because of the retrospective study design. We reviewed the data of 205 consecutive patients with traumatic CSCI who had undergone posterior decompression and fusion surgery in our institution between April 2017 and June 2021. Patients who met the following criteria were excluded: (1) American Spinal Injury Association Impairment Scale (AIS) grade E received at initial examination (n = 10); (2) follow-up period < 6 months (n = 36); (3) neurological status not evaluable because of brain injury, severe mental disorder, or other disturbance (n = 6). The remaining 153 patients who had undergone posterior decompression and fusion surgery were analyzed in this study.

Variables

Demographic and clinical variables for analysis included age, sex, height, body weight, body mass index (BMI), time to initial evaluation, comorbidities (ossification of the posterior longitudinal ligament, alcohol history, smoking history, hypertension, hyperlipidemia, ischemic heart disease, diabetes mellitus, malignancy, mental disease, and/or chronic renal dysfunction), mechanism of injury, and neurological status on admission. The time to initial evaluation was defined as the period from injury to the initial examination at our institution. Comorbidities were recorded from patient self-reports on admission. The mechanism of injury was divided into low- (e.g., a fall from a sitting or standing position) and high-impact trauma (e.g., a motor vehicle accident, a fall from more than 2 m, or a fall down a flight of stairs). Finally, neurological status on admission included AIS grade, neurological level of injury, and total motor index scores (MIS) of the upper and lower extremities. The motor index score uses a 0–5 scale for each key muscle, with a total of 25 points for each extremity [15].

Surgical data, including the surgical procedure, time to surgery, number of fusion levels, length of surgery, estimated blood loss (EBL), and use of perioperative transfusion, were obtained from the anesthetic and medical records. The time to surgery was defined as the time from injury to the start of the procedure.

Serum albumin and lymphocyte count were evaluated on admission, 3 days after injury, and 1, 2, 3, and 4 weeks after injury. The prognostic nutritional index (PNI) was calculated using the following formula: 0.005 × total lymphocyte count (/μL) + 10 × serum albumin concentration (g/dL) [16].

Perioperative complications included ventilator management, surgical site infection, pneumonia, urinary tract infection, cardiopulmonary dysfunction, deep venous thrombosis, and delirium within 30 days of surgery. A diagnosis of infectious complications was established by the Centers for Disease Control and Prevention [17]. According to microbiological criteria, symptoms of the surgical wound, respiratory infection, and/or systemic symptoms (fever, general discomfort) must be associated with bacteria for surgical site infection or pneumonia to be diagnosed. Cardiopulmonary dysfunction was defined as postoperative onset of symptomatic respiratory failure, pleural effusion, pulmonary edema, or acute respiratory disturbance [18]. Qualified sonographers used leg vein ultrasound to detect deep vein thrombosis at 3, 7, 14, and 28 days after surgery. Delirium was identified using a confusion evaluation approach based on medical data [19]. Perioperative complications were confirmed by the attending surgeons or other qualified medical personnel.

The cost of the acute-phase hospitalization period (within 90 days after surgery) was evaluated. All inpatient medical expenses for SCI surgery were extracted using medical fee data. Medical management or physical therapy charges were included in the total medical costs, along with hospital, surgery, and examination fees. Moreover, surgical costs comprised all expenses incurred during surgery, including those related to anesthetic administration and implants. Radiography, magnetic resonance imaging (MRI), computed tomography (CT), and blood sampling were all part of the examination cost. Finally, charges associated with hospital stay included perioperative centralized management, pharmaceutical therapy, and food costs.

Neurological outcomes at discharge included AIS grade and MIS score at discharge and improvements from admission to discharge as well as length of hospital stay. AIS improvement was defined as an improvement of one or more grades from admission to discharge, and MIS improvement was calculated as the difference between MIS at discharge and on admission.

Surgical indication

CT and MRI data were obtained for all the patients on admission before treatment. More than two physicians conducted clinical and radiological assessments and decided on the surgical indication, and one had more than five years of experience examining patients with spinal cord injuries. Physicians evaluated vertebral body or posterior element (facet, lamina, spinous process) fractures, disco–ligament complex (DLC) injuries, and facet joint dislocations with CT and MRI, and assessed instability of injuries based on morphology and DLC items in the SLIC score [20]. These characteristics were assessed along with neurological status, and a score of 5 or higher on the SLIC score was considered an indication for surgery, while a score of less than 4 was considered a conservative treatment [21].

The criteria for determining the number of fusion levels differed between the DISH and non-DISH groups. In the DISH group, as a rule, fixation was performed at 3 levels above and below as recommended in previous reports [5]. For the patients in the non-DISH group, the number of fusion levels was determined to be as few as possible based on bone quality, age, type of anchors, or injury morphology of the cervical spine.

Grouping

We assessed the CT images for the presence of DISH at the level of the SCI. The level of SCI was determined based on MRI images such as signal intensity of spinal cord and spinal cord edema. DISH was diagnosed according to the sagittal CT criteria defined as follows: (1) presence of bony bridge formation along the anterolateral aspects of at least four contiguous vertebral bodies; (2) relative preservation of intervertebral disc height; and (3) absence of apophyseal joint bony ankylosis and sacroiliac joint ankyloses [22]. Based on these criteria, patients were classified into DISH (presence of DISH fracture that caused spinal cord injury) and non-DISH (unstable CSCI without DISH at the level of SCI) groups.

Statistical analyses

Categorical variables were expressed as absolute numbers and percentages and analyzed using either Fisher’s exact test or a chi-square test. We used the Shapiro–Wilk test to determine whether continuous variables were normally distributed. These variables are expressed as mean ± standard deviation and were analyzed using unpaired t-tests. Statistical analyses were performed using SPSS 28.0 (IBM Corp., Armonk, NY, USA). Statistical significance was set at *p* < 0.05.

## 3. Results

Patient characteristics

The flowchart of the study selection criteria is shown in Figure 1. Among 153 patients, 24 (15.7%) were assigned to the DISH group and 129 to the non-DISH group. Of the DISH group, all patients had DISH fractures that caused spinal cord injury (Figure 2). Their baseline characteristics, traumatic impact, neurological status on admission, and surgical data are shown in Table 1. The DISH group had patients who were significantly older (*p* = 0.036), had a significantly higher proportion of men (*p* = 0.026), and had more patients with hypertension (*p* = 0.037) than the non-DISH group. Additionally, significantly more patients in the DISH group had low-impact trauma (*p* = 0.009) than their non-DISH counterparts. Height, body weight, BMI, and neurological status on admission did not significantly differ between the groups.

Furthermore, the DISH group had a significantly higher number of fusion levels (4.2 ± 1.5 vs. 1.8 ± 0.9, *p* < 0.001), a longer surgery length (200 ± 60 vs. 136 ± 44, *p* < 0.001), and a significantly higher EBL (438 ± 444 vs. 169 ± 212 mL, *p* = 0.008). Moreover, the DISH group more frequently received perioperative blood transfusions (41.7% vs. 11.4%, *p* < 0.001) compared with the non-DISH group.

Perioperative changes in laboratory data

Laboratory variables are shown in Table 2 and Figure 3. Patients in the DISH group compared with those in the non-DISH group had lower preoperative PNI scores on admission and reduced PNI and serum albumin levels at 3 and 4 weeks after surgery.

Clinical outcomes

Changes in the AIS grade from admission to discharge in the DISH and non-DISH groups are shown in Table 3 and Table 4, respectively, and other clinical outcomes are shown in Table 5. Regarding perioperative complications, the DISH group had significantly higher rates of ventilator management and pneumonia than the non-DISH group (*p* = 0.022 and 0.010, respectively). Moreover, medical costs were higher in the DISH group in the first month and 2 and 3 months after admission; however, these differences were not significant. No significant differences in AIS grade or MIS at discharge, MIS improvement, or length of hospital stay were observed between the groups.

## 4. Discussion

To our knowledge, this is the first study to compare the baseline characteristics, nutritional status, neurological prognosis, and perioperative complications between patients with CSCI with and without DISH. CSCI patients with DISH were older, more often male, and had poorer pre- and postoperative nutritional status. However, no difference was identified in neurological prognosis between patients with and without DISH. Nevertheless, patients with CSCI who also have DISH may be at a higher risk of developing respiratory complications.

The presence of DISH is associated with older age, male sex, obesity, hypertension, and diabetes mellitus [2,23]. Toyoda et al. reported a prevalence of 26%, which increased to 48% in patients over 70 years old [24]. Hirasawa et al. reported that DISH prevalence in men was approximately twice that in women [25]. Similarly, the present study showed that CSCI patients with DISH were older and more likely male than those without DISH. Significantly more patients with DISH had hypertension, whereas no significant difference in BMI or the prevalence of diabetes mellitus was observed between the two groups. Regarding trauma impact, spinal fractures tend to occur more frequently after minor trauma in patients with ankylosed spinal disorders (ASD) than in the general population. In a systematic review, Westerveld et al. reported that 66.3% of spinal fractures in patients with ASD were due to low-impact trauma [3]. Similarly, the proportion of low-impact trauma among patients in the DISH group in the present study was significantly higher than that in the non-DISH group.

While several studies have evaluated complications in patients with DISH undergoing surgical management for a spine fracture, perioperative changes in laboratory data have not been evaluated [4,6,7]. PNI can be easily calculated using parameters that are routinely measured in laboratory tests and has been widely used for nutritional assessment. Acarbas et al. reported that a preoperative PNI score below 47.7 was a significant risk factor for perioperative adverse events following spine surgery [16]. Because of the coefficient in the formula, the PNI value is affected to a greater extent by the serum albumin concentration. Hypoalbuminemia following SCI can be caused by a variety of factors, such as acute trauma, infection, surgical procedures, decreased hepatic synthesis, increased interstitial leakage, and rapid catabolism [26]. The present study demonstrated that PNI in patients with CSCI declined rapidly from injury to 3 days later and then gradually recovered; however it did not return to baseline even after 4 weeks. Considering the postinjury changes in PNI in this study, although the index may have decreased with surgical intervention, early surgery for patients with CSCI may be beneficial, not only for neurological prognosis [11] but also for the prevention of adverse events. The present study also revealed that patients with DISH had poorer nutritional status at initial evaluation than non-DISH patients. There are two possible reasons for this finding. First, patients with CSCI and DISH were significantly older than those without DISH. Serum albumin levels decrease with age [27], and this difference in age may have affected PNI. Second, while DISH is a noninflammatory disease, several recent reports have suggested that local chronic inflammation affects new bone formation in DISH [28,29]. Chronic inflammation depletes serum albumin in a wasting manner [14], which may also contribute to the low PNI in patients with CSCI and DISH. Furthermore, patients with DISH had lower PNI not only on admission but also at 3 and 4 weeks postoperatively. The results of this study indicate that patients with DISH have more invasive surgical procedures and a higher incidence of infectious complications such as pneumonia and urinary tract infections compared to non-DISH patients, which may contribute to a further decrease in PNI during hospitalization. Taking this information into consideration, it was suggested that nutritional intervention should begin earlier in the postoperative period for CSCI patients with DISH.

Reinhold et al. proposed that spinal columns in DISH show continuous bone-bridging hyperplasia on the exterior of the vertebral bodies, whereas the cancellous bone inside the vertebral body gradually deteriorates [30]. Additionally, the lever arm lengthens in spinal columns with several segments afflicted by ankylosing spondylitis. Therefore, Caron et al. recommended extending spinal fixation to three vertebrae above and below the fracture segments in patients with ASD [5]. Recent studies have shown that minimally invasive surgery, such as the percutaneous pedicle screw (PPS) technique, for spinal fractures in patients with an ankylosed spine decreased operation time, blood loss, and transfusion [31]. However, PPS fixation of the cervical spine is still not feasible, leaving CSCI patients with DISH with no choice but invasive long-term fixation. In fact, CSCI patients with DISH had a significantly higher number of fusion levels, a significantly longer length of surgery, a significantly higher EBL, and more frequent perioperative blood transfusions than the non-DISH group in the present study. Therefore, spine surgeons should keep in mind that patients with both CSCI and DISH require highly invasive surgical treatment despite their higher risk due to older age and poor nutritional status.

Complication rates and elevated mortality rates after surgical management have been reported to be higher in patients with fractures of ASD [4,6,7]. The complication rates of pneumonia and tracheostomy in patients with fractures of ASD are reported from 26.0% to 34.9% and 9.3 to 9.8%, respectively [3,5]. The complication rates of pneumonia and ventilator management in CSCI patients with DISH in this study were 29.2% and 16.7%, which were significantly higher than those in CSCI patients without DISH. According to Oudkerk et al., regardless of age, smoking habits, or BMI, people with DISH had lower FEV1% predicted, FVC% predicted, and lung capacity than those without DISH [32]. According to their theory, spinal ankylosis also affects the point of contact of the ribs, resulting in a rigid thoracic cage. In addition to these morphological factors, the present study revealed that patients with DISH are older and have poorer preoperative nutritional status, suggesting that these factors may influence perioperative complications.

In a systematic review with a population of 19,460, the proportion of patients who experienced at least one grade of improvement in AIS or the Frankel scale following traumatic spinal cord injury was reported at 49.4% [13]. The rate of neurological recovery in patients with SCI resulting from fractures of ASD ranges from 35.2% to 45.5% [3,4,7]. However, to our knowledge, no study has compared the neurological outcomes of CSCI between patients with and without DISH. In the present study, the rate of neurological recovery in CSCI patients with DISH was 54.2%, which was not statistically significantly different from the non-DISH group. Moreover, there was no significant difference in MIS at discharge and improvement in MIS, suggesting no differences in the neurological outcomes of patients with CSCI between those with and without DISH.

This study has several limitations. First, the sample size of patients with DISH was relatively small. Second, patients who underwent conservative treatment were excluded in order to evaluate perioperative complications. Therefore, the patient background may not reflect the factors of patients who underwent conservative treatment. Third, the retrospective data review from a single center limited our ability to deduce causal relationships. Fourth, while the DISH group had significantly higher rates of respiratory complications than the non-DISH group, we could not perform a regression analysis to figure out the causality because of the small number of respiratory complications. Finally, the possibility of selection bias in patient recruitment cannot be ruled out; however, we attempted to reduce this bias by selecting consecutive cases.

## 5. Conclusions

In conclusion, the present study suggests that no differences are found between the neurological outcomes of patients with CSCI with and without DISH. On the other hand, CSCI patients with DISH may require highly invasive surgery despite their older age and poor nutritional status and may be more likely to experience postoperative respiratory complications. We believe that these findings may provide physicians with a better understanding of neurological prognosis and perioperative complications in DISH accompanying CSCI, which they could impart to patients and their families.

## Figures and Tables

**Figure 1 jcm-12-05714-f001:**
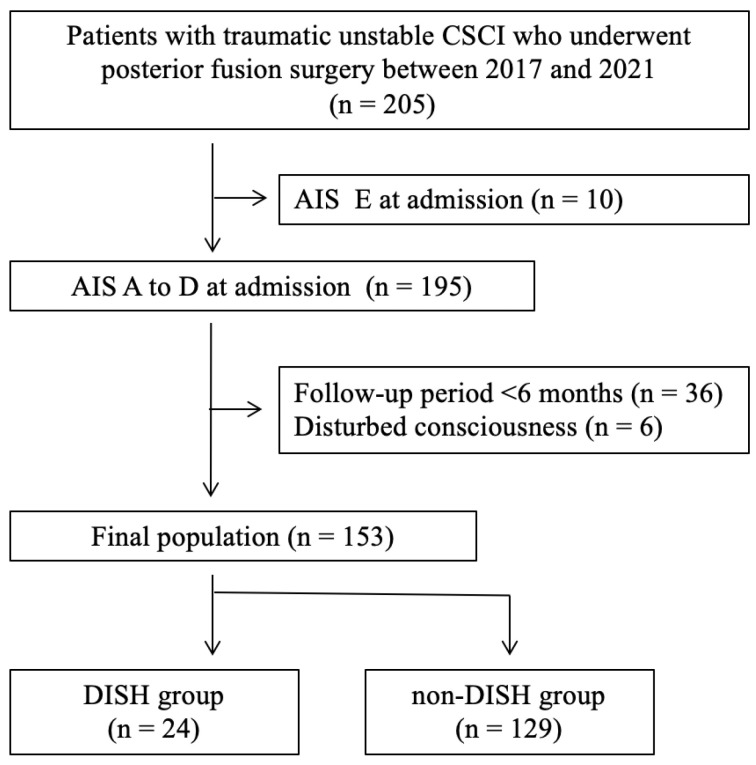
Flowchart of patient selection. AIS, American Spinal Injury Association Impairment Scale; CSCI, cervical spinal cord injury; DISH, diffuse idiopathic skeletal hyperostosis.

**Figure 2 jcm-12-05714-f002:**
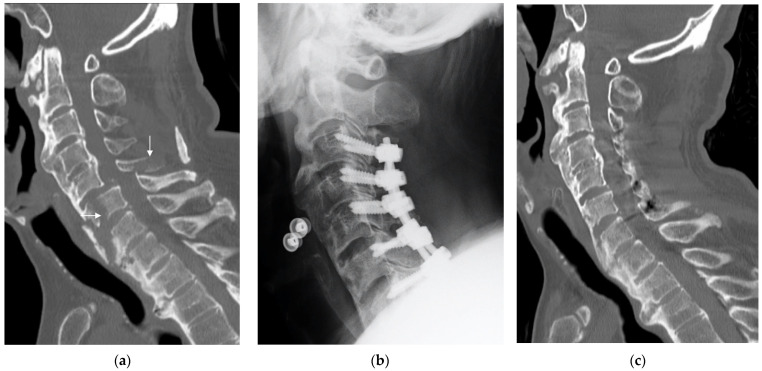
A representative case of cervical spinal cord injury with diffuse idiopathic skeletal hyperostosis. A 57-year-old man presented with a C5–6 three-column fracture after low-impact trauma. The preoperative AIS grade was B. We performed C3–7 posterior decompression and fusion surgery 6 h after the injury. Bone union was confirmed 3 months after surgery. One year after surgery, the patient recovered to AIS grade D. (**a**) Preoperative midsagittal computed tomography (CT) image with white arrows indicating the fracture line. (**b**) Radiograph 3 months after surgery. (**c**) Postoperative midsagittal CT 3 months after surgery (bone union confirmed).

**Figure 3 jcm-12-05714-f003:**
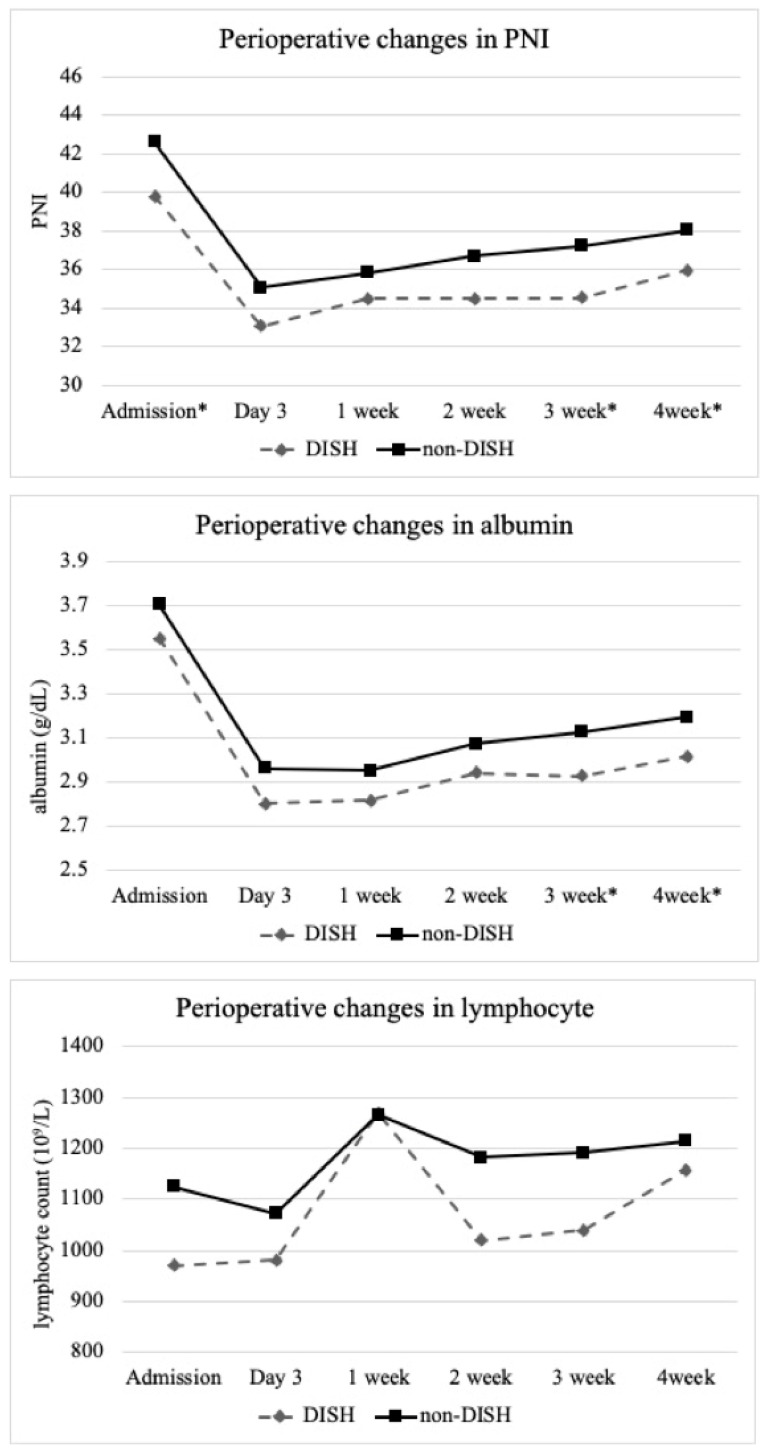
Perioperative changes in prognostic nutritional index (PNI), serum albumin, and lymphocyte count. DISH, diffuse idiopathic skeletal hyperostosis; * statistically significant difference (*p* < 0.05).

**Table 1 jcm-12-05714-t001:** Baseline characteristics and comparison of DISH and non-DISH groups.

Characteristic	Total	DISH Group n = 24	Non-DISH Group n = 129	*p* Value
Age (mean ± SD), years	67.0 ± 13.2	72.1 ± 9.7	65.9 ± 13.6	0.036 *
Sex male/female, n	130/23	24/0	106/23	0.026 *
Height (mean ± SD), cm	165.3 ± 8.4	166.8 ± 6.5	165.0 ± 8.8	0.336
Body weight (mean ± SD), kg	65.5 ± 14.3	70.4 ± 12.5	64.6 ± 14.6	0.069
BMI (mean ± SD), kg/m^2^	23.9 ± 4.3	25.2 ± 3.4	23.6 ± 4.4	0.099
Time to examination (mean ± SD), hours	20.8 ± 34.5	24.2 ± 37.3	20.2 ± 34.2	0.605
Comorbidities, n				
OPLL	47 (30.7%)	10 (41.7%)	37 (28.7%)	0.205
Alcohol	72 (45.8%)	12 (50.0%)	60 (45.0%)	0.663
Smoking	35 (22.9%)	3 (12.5%)	32 (24.8%)	0.188
Hypertension	68 (43.1%)	15 (62.5%)	51 (39.5%)	0.037 *
Hyperlipidemia	19 (12.4%)	4 (16.7%)	15 (11.6%)	0.503
Ischemic heart disease	12 (7.8%)	3 (12.5%)	9 (7.0%)	0.403
Diabetes mellitus	34 (22.2%)	7 (29.2%)	27 (20.9%)	0.373
Malignancy	27 (17.6%)	6 (25.0%)	21 (16.3%)	0.380
Mental disease	9 (5.9%)	1 (4.2%)	8 (6.2%)	1.000
Chronic renal disfunction	32 (20.9%)	5 (20.8%)	27 (20.9%)	0.991
Traumatic impact, n				0.009 *
Low	59 (38.6%)	15 (62.5%)	44 (34.1%)	
High	94 (61.4%)	9 (37.5%)	85 (65.9%)	
AIS grade on admission, n				0.352
A	37	7	30	
B	19	5	14	
C	54	8	46	
D	43	4	39	
NLI, n				0.309
C2	27	5	22	
C3	37	8	29	
C4	59	8	51	
C5	14	0	14	
C6	5	0	5	
C7	5	2	3	
C8	6	1	5	
MIS on admission, mean ± SD				
Upper extremity	19.6 ± 15.0	20.7 ± 16.7	19.4 ± 14.7	0.696
Lower extremity	17.7 ± 18.4	14.8 ± 18.8	183 ± 18.4	0.404
Total	37.3 ± 29.6	35.5 ± 29.8	37.6 ± 29.8	0.748
Time to surgery (mean ± SD), hours	22.8 ± 34.5	26.2 ± 37.3	22.2 ± 34.2	0.605
Number of fusion levels	2.2 ± 1.4	4.2 ± 1.5	1.8 ± 0.9	<0.001 *
Length of surgery (minutes)	146 ± 52	200 ± 60	136 ± 44	<0.001 *
Estimated blood loss (mL)	210 ± 277	438 ± 444	169 ± 212	0.008 *
Perioperative transfusion	25 (16.3%)	10 (41.7%)	15 (11.6%)	<0.001 *

* statistically significant difference. AIS, ASIA impairment scale; ASIA, American Spinal Injury Association; BMI, body mass index; DISH, diffuse idiopathic skeletal hyperostosis; MIS, motor index score; NLI, neurological level of injury; OPLL, ossification of the posterior longitudinal ligament; SD, standard deviation.

**Table 2 jcm-12-05714-t002:** Comparison of perioperative laboratory data between the DISH and non-DISH groups.

Laboratory Data, Mean ± SD	DISH Group n = 24	Non-DISH Group n = 129	*p* Value
Serum albumin at admission, g/dL	3.55 ± 0.43	3.70 ± 0.41	0.118
Serum albumin at day 3, g/dL	2.80 ± 0.28	2.96 ± 0.38	0.057
Serum albumin at 1 week, g/dL	2.82 ± 0.32	2.95 ± 0.43	0.156
Serum albumin at 2 weeks, g/dL	2.94 ± 0.40	3.07 ± 0.46	0.199
Serum albumin at 3 weeks, g/dL	2.93 ± 0.42	3.13 ± 0.46	0.049 *
Serum albumin at 4 weeks, g/dL	3.02 ± 0.37	3.20 ± 0.50	0.042 *
Lymphocyte at admission, count ×10^9^/L	971 ± 325	1127 ± 454	0.134
Lymphocyte at day 3, count ×10^9^/L	980 ± 378	1073 ± 396	0.300
Lymphocyte at 1 week, count ×10^9^/L	1269 ± 554	1270 ± 496	0.996
Lymphocyte at 2 weeks, count ×10^9^/L	1020 ± 305	1184 ± 414	0.068
Lymphocyte at 3 weeks, count ×10^9^/L	1040 ± 350	1198 ± 415	0.088
Lymphocyte at 4 weeks, count ×10^9^/L	1158 ± 329	1221 ± 360	0.423
PNI at admission	39.8 ± 4.6	42.5 ± 4.6	0.014 *
PNI at day 3	33.1 ± 3.4	35.1 ± 4.9	0.063
PNI at 1 week	34.5 ± 4.0	35.8 ± 5.3	0.257
PNI at 2 weeks	34.5 ± 4.4	36.7 ± 5.6	0.077
PNI at 3 weeks	34.5 ± 5.0	37.2 ± 5.6	0.031 *
PNI at 4 weeks	35.9 ± 4.1	38.1 ± 5.8	0.032 *

* statistically significant difference. PNI, prognostic nutritional index.

**Table 3 jcm-12-05714-t003:** Effect of changes in neurological status on AIS grade in DISH group.

	AIS Grade at Discharge					
	A	B	C	D	E	Total
AIS grade at admission						
A	6	1	0	0	0	7
B	0	1	1	3	0	5
C	0	0	1	7	0	8
D	0	0	0	3	1	4
Total	6	2	2	13	1	24

AIS, ASIA impairment scale; ASIA, American Spinal Injury Association; DISH, diffuse idiopathic skeletal hyperostosis.

**Table 4 jcm-12-05714-t004:** Effect of changes in neurological status on AIS grade in non-DISH group.

	AIS Grade at Discharge					
	A	B	C	D	E	Total
AIS grade at admission						
A	14	6	6	4	0	30
B	0	0	8	6	0	14
C	0	0	5	41	0	46
D	0	0	0	33	6	39
Total	14	6	19	84	6	129

AIS, ASIA impairment scale; ASIA, American Spinal Injury Association; DISH, diffuse idiopathic skeletal hyperostosis.

**Table 5 jcm-12-05714-t005:** Comparison of clinical outcomes between the DISH and non-DISH groups.

Perioperative Complications	DISH Group n = 24	Non-DISH Group n = 129	*p* Value
Ventilator management	4 (16.7%)	4 (3.1%)	0.022 *
Surgical site infection	0 (0%)	3 (2.3%)	1.000
Pneumonia	7 (29.2%)	11 (8.5%)	0.010 *
Urinary tract infection	10 (41.7%)	35 (27.1%)	0.151
Cardiopulmonary dysfunction	5 (20.9%)	25 (19.4%)	0.869
Deep venous thrombosis	7 (29.2%)	54 (41.9%)	0.244
Delirium	2 (8.3%)	16 (12.4%)	0.740
Medical costs (USD)			
First month after admission (A; n = 148)	2474 ± 878	2233 ± 568	0.217
First 2 months after admission (B, n = 134)	3232 ± 937	2937 ± 588	0.179
First 3 months after admission (C, n = 120)	3965 ± 1033	3569 ± 642	0.131
Second month (B-A; n = 134)	646 ± 251	677 ± 152	0.454
Third month (C-B; n = 120)	622 ± 118	613 ± 117	0.757
AIS grade at discharge, n			0.329
A	6 (25.0%)	14 (10.6%)	
B	2 (8.3%)	6 (4.5%)	
C	2 (8.3%)	19 (14.7%)	
D	13 (54.2%)	84 (65.1%)	
E	1 (4.2%)	6 (4.5%)	
AIS improvement (≥1 grade improvement)	13 (54.2%)	79 (61.2%)	0.516
MIS at discharge, mean ± SD			
Upper extremity	31.7 ± 18.8	32.9 ± 15.2	0.738
Lower extremity	25.7 ± 21.5	31.8 ± 19.4	0.170
Total	57.4 ± 37.9	65.2 ± 32.4	0.298
Improvement of MIS, mean ± SD			
Upper extremity	11.0 ± 14.1	13.9 ± 12.2	0.305
Lower extremity	10.9 ± 12.9	13.7 ± 16.4	0.431
Total	21.9 ± 25.1	27.5 ± 21.7	0.258
Length of hospital stay (days)	208 ± 101	202 ± 126	0.833

* statistically significant difference. AIS, ASIA impairment scale; ASIA, American Spinal Injury Association; DISH, diffuse idiopathic skeletal hyperostosis; MIS, motor index score; SD, standard deviation; USD, United States Dollar.

## Data Availability

The data analyzed during the current study are not publicly available due to their containing information that could compromise the privacy of research participants.

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
