# Peer review of "The Impact of Diffuse Idiopathic Skeletal Hyperostosis on Nutritional Status, Neurological Outcome, and Perioperative Complications in Patients with Cervical Spinal Cord Injury"

_jcm, 2023, doi:10.3390/jcm12175714_

Round 1

Reviewer 1 Report

We welcome the concept of the study comparting various outcomes after operative treatment of cervical spinal cord injury in patients with DISH to others.

There are some concerns regarding methods and definitions otherwise the paper is well written.

Line 53. The sentence »Evidently, both DISH fractures and SCI are associated with high complication rates.« is contradictory to the following text, when the authors describe that »the impact of DISH on CSCI patients has not yet been described«.

l. 77. define »time to initial evaluation«

l. 131. I assume that in all patients at least a CT has been performed before surgery. Please confirm.

l. 85, 132. How was determined the level of SCI? AIS? Were spine cord injuries only defined based on the AIS and MIS grade or were they or also demonstrated on the MRI? In how many patients the MRI has been performed to determine the presence and extend of the spinal canal pathology (spinal cord compression by the deformed vertebral body, posttraumatic hernia, epidural hematoma) and true spinal cord injury (oedema, bleeding, transsection?) Explain or mention it in the limitations.

l. 126. Indications for the surgery are described inconcistently, e.g. »dislocated fractures with SCI and cases of severe paralysis with severe stenosis« . Define severe paralysis and severe stenosis. Explain »conservative treatment was performed in principle for cases with less severe stenosis or less instability«. Define less severe stenosis and less instability. Otherwise this might represent a limitation.

l. 127, 128. How was defined the stability of injuries?

Author Response

Reviewer #1:

  1. Line 53. The sentence »Evidently, both DISH fractures and SCI are associated with high complication rates.« is contradictory to the following text, when the authors describe that »the impact of DISH on CSCI patients has not yet been described«.

Author’s Response: We sincerely thank you for your kind comment. As you pointed out, these two sentences contradict each other, so the following sentence has been deleted.

“Evidently, both DISH fractures and SCI are associated with high complication rates.”

  1. 77. define »time to initial evaluation«

Author’s Response: We are grateful for the reviewer’s important comment. Time to initial evaluation is already defined on line 80 as follows; “The time to initial evaluation was defined as the period from injury to the initial examination at our institution.” We hope you will confirm this.

  1. 131 I assume that in all patients at least a CT has been performed before surgery. Please confirm

Author’s Response: We are grateful for your important comment. As you noted, we performed CT and MRI on all our patients prior to treatment. Therefore, we have changed the sentence as follows.

“CT and MRI data were obtained for all the patients on admission before treatment.” (Methods section line 127)

  1. 85, 132. How was determined the level of SCI? AIS? Were spine cord injuries only defined based on the AIS and MIS grade or were they or also demonstrated on the MRI? In how many patients the MRI has been performed to determine the presence and extend of the spinal canal pathology (spinal cord compression by the deformed vertebral body, posttraumatic hernia, epidural hematoma) and true spinal cord injury (edema, bleeding, transsection?) Explain or mention it in the limitations.

Author’s Response: Thank you very much for your insightful suggestions. We performed MRI on all patients and determined the level of injury based on the MRI images. Therefore, we added the following sentence.

“The level of SCI was determined based on MRI images such as signal intensity of spinal cord and spinal cord edema.” (Methods section line 142-144)

  1. Indications for the surgery are described inconsistently, e.g. »dislocated fractures with SCI and cases of severe paralysis with severe stenosis« . Define severe paralysis and severe stenosis. Explain »conservative treatment was performed in principle for cases with less severe stenosis or less instability«. Define less severe stenosis and less instability. Otherwise this might represent a limitation. How was defined the stability of injuries?

Author’s Response: We are grateful for the reviewer’s important comment. I apologize for the confusion caused by the vague description of the indications for surgery. At our institution, instability of injuries and surgical indications are determined based on the SLIC score. Therefore, we have added the following sentence for more detailed surgical indications

“CT and MRI data were obtained for all the patients on admission before treatment. More than two physicians conducted clinical and radiological assessments and decided on the surgical indication and one had more than five years of experience examining patients with spinal cord injuries. Physicians evaluated vertebral body or posterior element (facet, lamina, spinous process) fractures, disk-ligament complex (DLC) injuries, and facet joint dislocations with CT and MRI, and assessed instability of injuries based on morphology and DLC items in the SLIC score. These characteristics were assessed along with neurological status, and a score of 5 or higher on the SLIC score was considered an indication for surgery, while a score of less than 4 was considered a conservative treatment.” (Methods section line 130-135)

Reviewer 2 Report

Dear Editor,

First of all, I would like to thank the authors for their work. The submission is well written and valuable, but some issues need re-consideration.

Based on the retrospective nature of the study, the protocol used by the authors is routine practice in their center? For instance, I mean that serum albumin and lymphocyte are checked on days 1 and 3, and 1,2, 3, and 4 weeks after trauma as a routine practice. And also, about ultrasonography for DVT?

The authors should explain the motor index score and total score for each extremity.

According to the results, in the patient characteristics section, there are no new data and it is found that DISH is more common in the elderly, in men, and minor trauma can significantly affect these patients, as in patients with ankylosing spondylitis. However, the results are valuable for neurologic outcomes and complications. Moreover, the authors evaluated the PNI score and showed that PNI in patients with DISH was lower on admission and decreased more during admission, and although they offered two explanations for its lower value on admission, the reasons for the further decrease in PNI during hospitalization have not been discussed. And the clinical significance of this finding is not mentioned. Did lower PNI only have a negative effect on complications? Is the higher risk of complications just due to PNI? If yes, this is true for all types of patients. Did they not analyze the effects of PNI on other variables? Did they not do a regression analysis for PNI effects?

Given the number of fusion levels, what does a level of 1.8 mean for the non-DISH group?

Yours sincerely,

Author Response

Reviewer #2:

  1. Based on the retrospective nature of the study, the protocol used by the authors is routine practice in their center? For instance, I mean that serum albumin and lymphocyte are checked on days 1 and 3, and 1,2, 3, and 4 weeks after trauma as a routine practice. And also, about ultrasonography for DVT?

Author’s Response: Thank you very much for your question. As you mentioned, our facility performs blood tests and ultrasound examinations at set intervals according to a protocol. Therefore, we have based our investigation on the results of the tests within that protocol.

  1. The authors should explain the motor index score and total score for each extremity.

Author’s Response: Thank you very much for your insightful suggestions. As you pointed out, we have added the following statement regarding the motor index score.

“The motor index score uses a 0–5 scale for each key muscle, with a total of 25 points for each extremity[15].” (Methods section line 86-87)

  1. According to the results, in the patient characteristics section, there are no new data and it is found that DISH is more common in the elderly, in men, and minor trauma can significantly affect these patients, as in patients with ankylosing spondylitis. However, the results are valuable for neurologic outcomes and complications. Moreover, the authors evaluated the PNI score and showed that PNI in patients with DISH was lower on admission and decreased more during admission, and although they offered two explanations for its lower value on admission, the reasons for the further decrease in PNI during hospitalization have not been discussed. And the clinical significance of this finding is not mentioned.

Author’s Response: We are grateful for the reviewer’s important comment. We found that patients in the DISH group had a lower PNI on admission and discussed two possible reasons for this. Moreover, patients with DISH had lower PNI not only on admission but also at 3 and 4 weeks postoperatively. From the results of the present study, we hypothesized that there were two reasons for this. First, CSCI patients with DISH required more invasive surgical procedures than those without DISH. Highly invasive surgery depleted serum albumin, which may have led to a further decline in PNI. Second, Second, patients in the DISH group had a higher incidence of infectious complications than those in the non-DISH group. The albumin depletion due to infection may be another cause of decline in PNI. Therefore, we added the following sentence in the Discussion section.

“Furthermore, patients with DISH had lower PNI not only on admission but also at 3 and 4 weeks postoperatively. The results of this study indicate that patients with DISH have more invasive surgical procedures and a higher incidence of infectious complications such as pneumonia and urinary tract infections compared to non-DISH patients, which may contribute to a further decrease in PNI during hospitalization. Taking these into consideration, it was suggested that nutritional intervention should begin earlier in the postoperative period for CSCI patients with DISH.” (Discussion section line 279-286)

Did lower PNI only have a negative effect on complications? Is the higher risk of complications just due to PNI? If yes, this is true for all types of patients. Did they not analyze the effects of PNI on other variables? Did they not do a regression analysis for PNI effects?

Author’s Response: Thank you very much for your question. In the present study, we found that the DISH group had significantly higher rates of respiratory complications than the non-DISH group. Respiratory complications could be caused by a variety of factors, including age, smoking history, and rigid thoracic cage. Furthermore, lower PNI is also a known risk factor for infectious complications. Therefore, it was desirable to analyze which of the following factors had the greatest impact on respiratory complications in the DISH group: older age, lower PNI, and a s rigid thoracic cage due to DISH. However, because of the small number of respiratory complications (ventilator management: 8 patients, pneumonia: 18 patients) in this study, we were unable to perform a regression analysis including PNI to figure out the causality. Therefore, we have added this to the limitation as follows.

“Fourth, while the DISH group had significantly higher rates of respiratory complications than the non-DISH group, we could not perform a regression analysis to figure out the causality because of the small number of respiratory complications.” (Discussion section line 331-333)

  1. Given the number of fusion levels, what does a level of 1.8 mean for the non-DISH group?

Author’s Response: Thank you very much for your question. In the non-DISH group, we determined the number of fusion levels as short as possible based on bone quality, age, type of anchors, or injury morphology of the cervical spine. On the other hand, fixation was usually performed at 3 levels above and below as recommended in previous reports for patients in the DISH group. We have added the following statement regarding the criteria for determining the number of fusion levels.

“The criteria for determining the number of fusion levels differed between the DISH and non-DISH groups. In the DISH group, as a rule, fixation was performed at 3 levels above and below as recommended in previous reports [5]. For the patients in the non-DISH group, the number of fusion levels was determined as short as possible based on bone quality, age, type of anchors, or injury morphology of the cervical spine.” (Methods section line 136-140)